# Development of Cell Permeable NanoBRET Probes for the Measurement of PLK1 Target Engagement in Live Cells

**DOI:** 10.3390/molecules28072950

**Published:** 2023-03-25

**Authors:** Xuan Yang, Jeffery L. Smith, Michael T. Beck, Jennifer M. Wilkinson, Ani Michaud, James D. Vasta, Matthew B. Robers, Timothy M. Willson

**Affiliations:** 1Structural Genomics Consortium, UNC Eshelman School of Pharmacy, University of North Carolina at Chapel Hill, Chapel Hill, NC 27599, USA; 2Promega Corporation, 2800 Woods Hollow Road, Madison, WI 53719, USAmatt.robers@promega.com (M.B.R.)

**Keywords:** kinase, assay, drug, in cell, target engagement

## Abstract

PLK1 is a protein kinase that regulates mitosis and is both an important oncology drug target and a potential antitarget of drugs for the DNA damage response pathway or anti-infective host kinases. To expand the range of live cell NanoBRET target engagement assays to include PLK1, we developed an energy transfer probe based on the anilino-tetrahydropteridine chemotype found in several selective PLK inhibitors. Probe **11** was used to configure NanoBRET target engagement assays for PLK1, PLK2, and PLK3 and measure the potency of several known PLK inhibitors. In-cell target engagement for PLK1 was in good agreement with the reported cellular potency for the inhibition of cell proliferation. Probe **11** enabled the investigation of the promiscuity of adavosertib, which had been described as a dual PLK1/WEE1 inhibitor in biochemical assays. Live cell target engagement analysis of adavosertib via NanoBRET demonstrated PLK activity at micromolar concentrations but only selective engagement of WEE1 at clinically relevant doses.

## 1. Introduction

The polo-like kinases are Ser/Thr protein kinases that play key roles in the cell cycle and mitosis and are often dysregulated in cancer [1,2]. Five paralogs are found in humans, PLK1–5, of which PLK1 is the most highly studied member. The inhibition of PLK1 results in aberrant chromosome segregation, mitotic block, and cell death [2]. Notably, PLK2, PLK3, and PLK4 have also been implicated in various aspects of cell cycle control [3,4]. PLK5 lacks a functional kinase domain and is included in the family only because of homology in its other domains. Many ATP-competitive PLK inhibitors have been characterized in biochemical enzyme inhibition assays, with inhibitors of PLK1 often showing activity on PLK2 and PLK3 but little or no activity on PLK4. Conversely, PLK4 inhibitors generally do not show activity on PLK1–3. Although PLK1 inhibitors show robust anticancer activity in animal models, they have been plagued by dose-limiting toxicity in clinical trials and none have been approved as cancer drugs to date [1]. BI2536 was the first PLK1 inhibitor to enter clinical trials. Both BI2536 and its follow-on analog volasertib (Figure 1) were reported to be nanomolar inhibitors of PLK1–3 [5,6]. Later generation compounds such as GSK461364 [7] and onvansertib [8] were reported as being nanomolar inhibitors with selectivity for PLK1 over PLK2 and PLK3. It remains unclear, however, whether pan-PLK or selective PLK1 inhibition will be clinically superior for treating solid tumors, since there is often a disconnect between the potency of enzyme inhibition and the high micromolar doses required to see therapeutic effects [2].

Beyond their established role as cancer targets, protein kinases have emerged as promising host cell targets for many infectious diseases. For example, viruses often co-opt host kinases to affect cell entry and replication [9,10]. Protein kinases also play important roles in the host immune response to viral and bacterial infection, such that kinase inhibitors are being identified for use in combination with conventional anti-infective drugs [11,12]. In addition, the suppression of the inflammatory pathways with JAK/STAT inhibitors can aid the recovery of patients in the post-infection phase of the disease [9]. PLK1 has also been reported as a collateral target of inhibitors of WEE1 kinase, a synthetic lethality target of the DNA damage response [13]. In all of these cases, PLK1 can be considered as an antitarget since its inhibition results in potential pathway antagonism or can confound the optimization of host cell kinase inhibitors for infectious disease therapy due to cell toxicity [14].

We have shown that live cell NanoBRET target engagement assays can be used to measure the cellular potency of protein kinase inhibitors [15,16]. These NanoBRET assays use promiscuous kinase inhibitors conjugated to BODIPY 576/589 as cell penetrant probes (also known as tracers) to generate bioluminescence resonance energy transfer (BRET) when combined with ultrabright NanoLuc (NLuc)-kinase fusion proteins [17]. Competition of the fluorescent tracer by ATP-competitive kinase inhibitors gives a measure of the in-cell target engagement that often correlates with the cellular potency of the inhibitors determined by more time-consuming phospho-substrate assays [15]. Unfortunately, none of the promiscuous tracers that had been previously developed were shown to work with PLK1 [15]. Therefore, to address this critical gap in the NanoBRET probe set, we developed a specific tracer for this important signaling kinase starting from a known PLK inhibitor chemotype.

## 2. Results

### 2.1. Synthesis of Fluorescence Energy Transfer Probes ***10*** and ***11***

Bifunctional PLK1 energy transfer probes were developed from the anilino-tetrahydropteridine core present in the ATP-competitive PLK inhibitors BI 2536 and volasertib [5,6] (Figure 1). Analysis of the PLK1 crystal structure (PDB: 2RKU) suggested that the installation of a linker on the piperidine ring of intermediate **1** would be optimal for the placement of the BODIPY 576/589 fluorophore. The synthesis of intermediate **1** started by HATU-mediated coupling between tert-butyl 4-aminopiperidine-1-carboxylate (**2**) and 3-methoxy-4-nitrobenzoic acid (**3**) followed by the reduction of product **4** using hydrazine hydrate solution catalyzed by Raney-Nickel to yield the known aniline **5** [18] (Scheme I). The coupling of aniline **5** with the commercially available tetrahydropteridine **6** using a Buchwald–Hartwig amination gave compound **7**, which was *N*-Boc deprotected under acidic conditions to yield the key intermediate **1**. Two potential energy transfer probes were synthesized from intermediate **1**, first via the coupling of intermediate **1** to BODIPY 576/589 NHS ester (**9**) in the presence of DIPEA to yield probe **10**. Alternatively, a tetraethylene glycol (PEG)_4_ linker was attached to intermediate **1** followed by coupling to BODIPY 576/589 to yield energy transfer probe **11** (Figure 1).

### 2.2. Development of a PLK1 NanoBRET Assay

We conducted the probe titration studies in live HEK293 cells in the adherent (ADH) format [15] via energy transfer using plasmid DNAs encoding *N*- and *C*-terminal fusions of full-length PLK1 with NLuc. We found that for both probes **10** and **11**, NLuc that was fused to the *N*-terminus of human PLK1 (NLuc-PLK1) gave a much stronger BRET signal (BRET ratio ~39 for probe **10** and ~48 for probe **11**) compared to NLuc fused to the C-terminus of PLK1 (PLK1-NLuc) (BRET ratios ~15 for probe **10** and **11**) (Figure 2a). Although both probes gave good BRET ratios using NLuc-PLK1, the signal using probe **11** with the (PEG)_4_ linker was consistently larger than with probe **10**. Thus, we selected probe **11** for further characterization and optimization of the PLK1 NanoBRET assay. The apparent affinity of intracellular energy transfer probe **11** was measured by treating the cells with increasing concentrations in the presence or absence of an excess (20 μM) of unlabeled intermediate **1** as a competitive PLK1 inhibitor (Figure 2b). The BRET ratio was plotted versus NanoBRET energy transfer probe **11** concentration to determine an apparent intracellular potency, EC_50_ = 0.30 μM. To optimize concentrations for the PLK1 NanoBRET assay, the apparent cellular affinity of the unlabeled intermediate **1** was measured at multiple fixed concentrations of the energy transfer probe **11** (Figure 2c) with the goal of selecting a concentration that was at or below the EC_50_ of the probe for the target protein, as prior results have demonstrated that lower probe concentrations resulted in the more accurate estimation of intracellular compound affinity.

A concentration of 0.2 μM of energy transfer probe **11** was selected as the optimal concentration for the PLK1 NanoBRET assay as it yielded a high assay window (>3 fold) and was below the EC_50_ for the probe. Replotting the apparent IC_50_ values for unlabeled intermediate **1** against the concentration of NanoBRET energy transfer probe **11**, a linearized Cheng–Prusoff analysis yielded an apparent K_D_ of 0.12 μM for unlabeled compound **1** for the intracellular target engagement of PLK1 (Figure 2d). To test whether energy transfer probes **10** or **11** could also be used to develop NanoBRET assays for PLK2 and PLK3, the BRET signal was measured using NLuc fused to the *N*-terminal of human PLK2 (NLuc-PLK2) or human PLK3 (NLuc-PLK3). As shown in Figure 2a, robust signals with BRET ratios of 24–38 were obtained using both probes. Since the PLK1 NanoBRET assay had been optimized using probe **11**, we selected the same probe for the development of PLK2 and PLK3 NanoBRET assays using the identical procedures. The optimized concentration of energy transfer probe **11** for the NanoBRET assays was 1 μM with both NLuc-PLK2 and NLuc-PLK3 (Appendix A), which gave assay windows of 2.4- and 2.1-fold, respectively.

### 2.3. In-Cell Target Engagement of PLK Inhibitors

Having developed NanoBRET assays for PLK1–3 using energy transfer probe **11**, we selected a series of known PLK inhibitors to compare their enzyme inhibition potency with intracellular target engagement. Several factors can affect the cellular potency of ATP-competitive kinase inhibitors, including their ability to cross cell membranes and the K_m_(ATP) of the individual kinase, which determines the effect of intracellular ATP as a competitor. The IC_50_s for seven PLK inhibitors were determined in the PLK1, PLK2, and PLK3 NanoBRET assays in HEK293 cells. As shown in Table 1, all seven inhibitors demonstrated potent intracellular target engagement on PLK1 and some showed activity on PLK2 and PLK3. The NanoBRET/Enzyme IC_50_ values showed that many of the PLK1 inhibitors were less potent in cells, although the ratio differed for each inhibitor (Table 1). Notably, the potency of the PLK1 in-cell target engagement was in good agreement with the lowest concentration for the inhibition of cell proliferation, as reported in the literature for each inhibitor using a range of cell lines.

### 2.4. In-Cell Target Selectivity of Adavosertib

To further demonstrate the utility of the PLK1 NanoBRET assay, we profiled adavosertib (MK-1775/AZD1775), a WEE1 kinase inhibitor in clinical development for the treatment of pancreatic cancer, whose kinase selectivity has been the subject of debate in the literature [13,22,23]. 

In the WEE1 NanoBRET assay using the K10 tracer [24], adavosertib demonstrated an IC_50_ = 17 nM, which is in agreement with the results of the inhibition of substrate phosphorylation in cells [23]. In contrast, using the energy transfer probe **11** in the PLK1 NanoBRET assay, adavosertib showed >50% target occupancy only at micromolar concentrations above its reported circulating plasma levels in patients using a 225mg dose [25,26] (Figure 3a). To further annotate the in-cell selectivity of adavosertib, we utilized our broad-spectrum profiling protocol employing the energy transfer probe K10, which enables the assay of 192 kinases but not PLK1 using a single tracer [24]. At a concentration of 1 µM, adavosertib showed >50% target engagement of only seven kinases (Figure 3b). Notably, at this concentration, adavosertib showed 95% and 53% target engagement of PLK2 and PLK3, respectively.

## 3. Discussion

We have developed an energy transfer probe **11** that can be used to configure NanoBRET assays for PLK1, PLK2, and PLK3, addressing a critical gap in PLK family target engagement profiling. The optimized PLK1 NanoBRET assay allows, for the first time, researchers to determine the intracellular target engagement of PLK1 inhibitors, which further expands the utility of this assay technology to support the lead optimization and counter screening of this important drug target.

Since the cellular concentration of ATP (usually 1–10 mM) is often much higher than the biochemical K_m_(ATP) of kinase enzymes, ATP-competitive kinase inhibitors are often less potent in intact cells than predicted from the cell-free assays, and it is important to determine intracellular target engagement to be able to interpret their phenotypic effects [27]. For PLK1, the biochemical K_m_(ATP) is 2–5 μM [28]. By screening a series of PLK inhibitors in preclinical and clinical development, we demonstrated that their in-cell target engagement was often lower than predicted by their biochemical enzyme inhibition potency (Table 1). While Ro3280 showed no drop in cellular potency, volasertib was >20-fold less potent in cells than its enzyme potency. The other PLK inhibitors showed a loss in cell potency ranging from 5- to 11-fold. Each of the seven PLK1 inhibitors have been reported to show cytotoxic activity in a range of cancer cell lines. It is notable that the lowest IC_50_ values for the inhibition of cell proliferation closely matched the IC_50_ for intracellular target engagement of PLK1 via NanoBRET for all seven inhibitors (Table 1). BI2536 and volasertib, first and second generation PLK inhibitors in the anilino-tetrahydropteridine chemotype, were also active in the PLK2 and PLK3 NanoBRET assays matching their pan-PLK enzyme inhibition profile. TPKI-26, another derivative of the anilino-tetrahydropteridine chemotype [19], was a dual PLK1/PLK2 inhibitor in the NanoBRET assay but showed only weak activity on PLK3. Ro3280, which was published as a selective PLK1 inhibitor [20], was profiled as a pan-PLK inhibitor in cells, but with a preference for PLK1 > PLK2 > PLK3. GSK461324, a time-dependent PLK inhibitor with a complex selectivity profile assessed using FRET-based biochemical assays [7], was shown to be highly PLK1-selective when profiled in the live cell NanoBRET target engagement assay. The third generation inhibitor MLN0905 was also selective for PLK1 in the NanoBRET target engagement assays, although it showed weak activity on both PLK2 and PLK3, which is consistent with its binding to both of these kinases at 1 µM in an Ambit KINOMEscan assay [21]. Finally, the third generation PLK1 inhibitor onvansertib was highly PLK1-selective in the NanoBRET assay with an IC_50_ on PLK1 that matched its potency for the inhibition of cell proliferation [8]. 

The PLK NanoBRET assays use a promoterless plasmid to maintain low expression levels of the kinase-NLuc fusions, which has been demonstrated not to impact the observed target occupancy in related assays [15,16]. In addition, while it has been reported that PLK1 is not catalytically active at all stages of the cell cycle [29], the activation state of the enzyme did not appear to impact the measurement of inhibitor target engagement via NanoBRET assay. Although it is possible that PLK1 inhibitors could impact the viability of HEK293 cells within the 2h incubation period of the assay, an extracellular NLuc inhibitor [30] was used to block the BRET donor signal from cells that had compromised membranes, ensuring that only intracellular target engagement was measured in the assay. Our results demonstrate that these NanoBRET assays allow, for the first time, direct comparison of the intracellular potency and isozyme selectivity of PLK inhibitors under development as cancer drugs. Ideally, these assays would be conducted in a cancer cell lineage that reflects the target tissue of the drug. However, there is often difficulty in transfecting these cells and controlling the expression level of the kinase-NLuc fusion. Instead, we chose to develop an optimized NanoBRET assay in HEK293 cells as a bridge between cell-free biochemical assays and assays performed in patient tumor cells.

The utility of the PLK1 NanoBRET assay to support the profiling of a kinase inhibitor of clinical importance was applied to the drug adavosertib, which was originally reported as being a selective WEE1 inhibitor [22] but was subsequently described as a dual WEE1/PLK1 inhibitor on the basis of chemical proteomic and biochemical enzyme inhibition assays [13]. A more recent report suggested that adavosertib was a selective WEE1 inhibitor in cells at clinically relevant doses by profiling the phosphorylation of downstream substrates using immunoblots [23]. These conflicting results opened the possibility that either the biochemical analysis of PLK1 inhibition may not translate in vivo, or that protein phosphorylation analysis may not accurately capture specific PLK1 activity in cells. Using the PLK1 and WEE1 NanoBRET assays combined with broad profiling across 192 kinases, we demonstrated that adavosertib was indeed a selective WEE1 inhibitor in cells at the low micromolar concentrations achieved using a 225 mg clinical dose, but did also exhibit off-target PLK activity with the rank order of PLK2 > PLK3 ~ PLK1 at higher micromolar concentrations that are reached with increased drug dosage or in the presence of a CYP3A4 inhibitor [31].

In summary, we have developed energy transfer probes to enable PLK NanoBRET assays that further expand the utility of this technology to profile the in-cell target engagement of kinase inhibitors. These NanoBRET assays provide a robust and rapid method to determine the cellular potency and isozyme selectivity of PLK inhibitors, which can aid lead optimization and counter screening for undesirable off-target activity.

## 4. Materials and Methods

### 4.1. Chemistry

#### 4.1.1. General Methods

The kinase inhibitors BI2536, volasertib, Ro3280, GSK461364, MLN0905, and adavosertib were purchased from MedChemExpress. TPKI-26 was donated by Takeda Pharmaceuticals as part of the Kinase Chemogenomic Set (KCGS) [14]. All chemical reagents were commercially available except those whose synthesis is described below. All reaction mixtures and column eluents were monitored via analytical thin-layer chromatography (TLC) performed on pre-coated Sorbtech fluorescent silica gel plates, 200 μm with an F254 indicator; visualization was accomplished by UV light (254/365 nm). Column chromatography was undertaken with a Biotage Isolera One instrument. Nuclear magnetic resonance (NMR) spectrometry was run using a Varian Inova 400 MHz or Bruker Avance III HD 850 MHz spectrometer. The NMR data were processed using MNova 14.3.1 (Mestrelab, Escondido CA, USA). Chemical shifts are reported in ppm with residual solvent peaks referenced as the internal standard. Analytical LC/MS data were obtained using a Waters Acquity ultrahigh-performance liquid chromatography (UPLC) system equipped with a photodiode array (PDA) detector running an acetonitrile/water gradient. Samples for high-resolution mass spectrometry (HRMS) were analyzed using a Q Exactive HF-X (ThermoFisher, Bremen, Germany) mass spectrometer. Samples were introduced by a heated electrospray source (HESI) at a flow rate of 10 µL/min. One hundred time domain transients were averaged in the mass spectrum. HESI source conditions were set as follows: nebulizer temperature 100 °C, sheath gas (N_2_) 15 arb, auxiliary gas (N_2_) 5 arb, sweep gas (N_2_) 0 arb, capillary temperature 250 °C, RF 100 V. The mass range was set to 200–2000 *m/z*. All measurements were recorded at a resolution setting of 120,000.

#### 4.1.2. Synthesis of Florescence Energy Transfer Probes **10** and **11**

##### Tert-butyl 4-(3-methoxy-4-nitrobenzamido)piperidine-1-carboxylate (**4**) 

To a solution of 3-methoxy-4-nitrobenzoic acid (**3**) (1.05 g, 1 eq, 5.32 mmol) in DMF (12 mL), HATU (2.56 g, 1.5 eq, 7.98 mmol) and DIPEA (1.38 g, 1.9 mL, 2 eq, 10.64 mmol) were added. The reaction mixture was stirred for 10 min at room temperature before adding tert-butyl 4-aminopiperidine-1-carboxylate (**2**) (1.06 g, 1 eq, 5.32 mmol). After 12 h, the reaction mixture was concentrated, and the resulting material was purified via chromatography on silica gel (eluent of ethyl acetate/hexane = 0−40%) to give the impure product. The desired product was recrystallized in the test tubes (40% ethyl acetate in hexanes). The white crystals were filtered and dried under vacuum to afford the desired product (**4**) (1.40 g, 69%). TLC condition: hexanes/ethyl acetate = 1/1. ^1^H NMR (400 MHz, CD_3_OD) δ 7.86 (d, *J* = 8.2 Hz, 1H), 7.69 (s, 1H), 7.50 (d, *J* = 8.9 Hz, 1H), 4.18–4.04 (m, 3H), 4.03 (s, 3H), 3.02–2.85 (m, 2H), 2.01–1.93 (m, 2H), 1.59–1.49 (m, 2H), 1.49–1.46 (m, 9H). ^13^C NMR (101 MHz, CD_3_OD) δ 167.5, 156.5, 153.6, 143.0, 140.9, 126.1, 120.3, 114.0, 81.2, 57.3, 49.1, 32.5, 28.7.

##### Tert-butyl 4-(4-amino-3-methoxybenzamido)piperidine-1-carboxylate (**5**) 

To a suspension of tert-butyl 4-(3-methoxy-4-nitrobenzamido)piperidine-1-carboxylate (**4**) (900 mg, 1 eq, 2.37 mmol) in ethanol (20 mL), hydrazine hydrate solution (50−60%) (1.7 mL) was added and the reaction mixture was stirred at 50 °C for 15 min until the starting material was dissolved completely. Excess of Raney-Nickel (1.39 g, 10 Eq, 23.7 mmol) was added to the reaction mixture, and the reaction was maintained at 50 °C. After 1 h, when gas evolution ceased, Raney-Nickel was removed via filtration under vacuum. The filtrate was concentrated in vacuo to give the desired product (**5**) (806 mg, 97%). TLC condition: hexanes/ethyl acetate = 2/1. ^1^H NMR (400 MHz, CD_3_OD) δ 7.36–7.26 (m, 2H), 6.73–6.67 (m, 1H), 4.11 (d, *J* = 13.8 Hz, 2H), 4.06–3.99 (m, 1H), 3.89 (s, 3H), 3.01–2.79 (m, 2H), 1.91 (dd, *J* = 12.6, 3.4 Hz, 2H), 1.57–1.43 (m, 11H). ^13^C NMR (101 MHz, CD_3_OD) δ 169.8, 156.5, 147.8, 142.5, 123.8, 122.3, 114.2, 110.5, 81.1, 56.1, 48.6, 32.8, 28.7. LC-MS (ESI) *m/z* calc. for C_17_H_28_N_3_O_4_ [M + H]^+^: 350.2, found 350.1.

##### (*R*)-4-((7-ethyl-8-isopropyl-5-methyl-6-oxo-5,6,7,8-tetrahydropteridin-2-yl)amino)-3-methoxy-N-(piperidin-4-yl)benzamide (**1**)

Tert-butyl 4-(4-amino-3-methoxybenzamido)piperidine-1-carboxylate (**5**) (600 mg, 1 eq, 1.72 mmol), (*R*)-2-chloro-7-ethyl-8-isopropyl-7,8-dihydropteridin-6(5*H*)-one (**6**) (437 mg, 1 eq, 1.72 mmol), 2-(dicyclohexylphosphanyl)-2’,4’,6’-tris(isopropyl)biphenyl (XPhos) (164 mg, 0.2 eq, 343 µmol), K_2_CO_3_ (949 mg, 4 eq, 6.87 mmol), and Pd_2_(dba)_3_ (157 mg, 0.1 eq, 172 µmol) were added into t-BuOH (15 mL) under Ar atmosphere. The reaction mixture was degassed and purged with Ar and then heated in a microwave at 120 °C for 30 min. The mixture was filtered through Celite, washed with DCM, and concentrated. The resulting material was purified via chromatography on silica gel (methanol/DCM = 0–10%) to give the crude product (**7**), 0.91 g, as a light yellow solid, which was used directly in the next step. TLC condition: DCM/MeOH = 10/1. LC-MS (ESI) *m/z* calc. for C_30_H_44_N_7_O_5_ [M + H]^+^: 582.3, found 582.2. 

To a solution of crude compound **7** (300 mg) in DCM (3 mL), TFA (3 mL) was added at room temperature. The reaction mixture was stirred for 1 h at room temperature. The reaction solution was concentrated to give a crude product which was purified via reverse phase chromatography on a C18 column (0–20% ACN in water + 0.1% TFA) to afford the desired product (**1**) as trifluoroacetate salt (193 mg, 332 μmol, 59% for two steps). Product **1** was converted to an HCl salt by adding 1M HCl in diethyl ether, and it was concentrated. TLC condition: DCM/MeOH = 10/1+ 5% NH_4_OH. ^1^H NMR (850 MHz, CD_3_OD) δ 8.02 (d, *J* = 7.6 Hz, 1H), 7.66 (s, 1H), 7.65 (s, 1H), 7.61 (d, *J* = 7.5 Hz, 1H), 4.63–4.54 (m, 2H), 4.24–4.19 (m, 1H), 4.03 (s, 3H), 3.52 (d, *J* = 12.5 Hz, 2H), 3.32 (s, 3H), 3.21–3.15 (m, 2H), 2.24–2.20 (m, 2H), 2.14–2.08 (m, 1H), 2.01–1.94 (m, 3H), 1.48 (dd, *J* = 16.1, 6.2 Hz, 6H), 0.88 (t, *J* = 7.2 Hz, 3H). ^13^C NMR (214 MHz, CD_3_OD) δ 168.8, 164.5, 154.4, 152.4, 150.4, 133.0, 129.8, 124.1, 123.8, 121.2, 117.8, 111.6, 61.2, 56.9, 53.3, 49.5, 49.4, 46.5, 44.5, 29.5, 28.9, 28.9, 20.6, 19.5, 8.4. LC-HRMS (ESI) *m/z* calc. for C_25_H_36_N_7_O_3_ [M + H]^+^: 482.28741, found 482.28751.

##### (*R*)-*N*-(1-(3-(5,5-difluoro-7-(1H-pyrrol-2-yl)-5H-5^λ^,6^λ^-dipyrrolo[1,2-c:2’,1’-*f*][1-3]diazaborinin-3-yl)propanoyl)piperidin-4-yl)-4-((7-ethyl-8-isopropyl-5-methyl-6-oxo-5,6,7,8-tetrahydropteridin-2-yl)amino)-3-methoxybenzamide (**10**)

(*R*)-4-(4-((7-ethyl-8-isopropyl-5-methyl-6-oxo-5,6,7,8-tetrahydropteridin-2-yl)amino)-3-methoxybenzamido)piperidin-1-ium, trifluoracetate (**1**) (11 mg, 1 eq, 18 μmol) was charged into a flask and was taken up in anhydrous DMF (0.5 mL). To the stirred solution, DIPEA (12 mg, 16 μL, 5 eq, 92 μmol) was added, followed by BODIPY 576/589 NHS ester (**9**) (7.9 mg, 1 eq, 18 μmol). The mixture was stirred for 1 h and then subjected to preparative HPLC (column: Phenomenex, Luna 5 μM Phenyl-Hexyl, 100 Å, 75 × 30 mm, 5 micron; mobile phase A: water (0.05% TFA), B: methanol; method: 20–100% B, 6 min + 100% B, 4 min). The desired product (**10**) was obtained as a dark purple solid (10 mg, 13 μmol, 68%). TLC condition: DCM/MeOH=10/1. ^1^H NMR (400 MHz, CD_3_OD) δ 7.98 (d, *J* = 8.4 Hz, 1H), 7.59 (s, 1H), 7.58 (d, *J* = 1.9 Hz, 1H), 7.53 (dd, *J* = 8.4, 1.9 Hz, 1H), 7.23 (s, 1H), 7.22–7.15 (m, 3H), 7.00 (d, *J* = 4.6 Hz, 1H), 6.92 (d, *J* = 3.9 Hz, 1H), 6.37–6.30 (m, 2H), 4.64–4.47 (m, 3H), 4.21–4.05 (m, 2H), 3.98 (s, 3H), 3.29–3.14 (m, 6H), 2.88–2.78 (m, 3H), 2.12–1.83 (m, 4H), 1.63–1.47 (m, 2H), 1.42 (dd, *J* = 9.2, 6.8 Hz, 6H), 0.84 (t, *J* = 7.4 Hz, 3H). ^13^C NMR (214 MHz, CD_3_OD) δ 172.7, 168.6, 164.5, 156.2, 154.4, 152.3, 152.2, 150.5, 139.0, 135.0, 133.3, 133.2, 129.8, 127.4, 126.8, 124.9, 124.7, 124.2, 123.5, 121.1, 121.0, 119.0, 117.7, 117.6, 112.4, 111.5, 61.1, 56.7, 53.2, 49.9 (2C), 46.0, 43.8, 42.3, 33.8, 28.8, 28.7, 25.8, 20.5, 19.4, 8.3. LC-HRMS (ESI) *m/z* calc. for C_41_H_48_BF_2_N_10_O_4_ [M + H]^+^: 793.39156, found 793.39227.

##### (*R*)-*N*-(1-(1-amino-3,6,9,12-tetraoxapentadecan-15-oyl)piperidin-4-yl)-4-((7-ethyl-8-isopropyl-5-methyl-6-oxo-5,6,7,8-tetrahydropteridin-2-yl)amino)-3-methoxybenzamide (**8**)

To a solution of 2,2-dimethyl-4-oxo-3,8,11,14,17-pentaoxa-5-azaicosan-20-oic acid (t-Boc-N-amido-PEG4-acid) (38 mg, 1 eq, 0.10 mmol) in DMF (2 mL), TBTU (50 mg, 1.5 eq, 0.16 mmol) and *N*-ethyl-*N*-isopropylpropan-2-amine (54 mg, 4 eq, 0.42 mmol) were added and the mixture was stirred for 15 min. (*R*)-4-(4-((7-ethyl-8-isopropyl-5-methyl-6-oxo-5,6,7,8-tetrahydropteridin-2-yl)amino)-3-me-thoxybenzamido)piperidin-1-ium, trifluoracetate (**1**) (62 mg, 1 Eq, 0.10 mmol) was added to the reaction mixture and stirred for 18 h. Water was added to the reaction mixture and it was extracted with ethyl acetate. The crude product was purified via column chromatography on silica gel (0–10% MeOH in DCM) to afford tert-butyl (*R*)-(15-(4-(4-((7-ethyl-8-isopropyl-5-methyl-6-oxo-5,6,7,8-tetrahydropteridin-2-yl)amino)-3-methoxybenzamido)piperidin-1-yl)-15-oxo-3,6,9,12-tetraoxapentadecyl)carbamate (50 mg, 60 μmol, 58%). TLC condition: 10% MeOH in DCM. ^1^H NMR (850 MHz, CD_3_OD) δ 8.52 (d, 1H), 7.75 (s, 1H), 7.50–7.47 (m, 2H), 4.69 (hept, *J* = 6.9 Hz, 1H), 4.60–4.55 (m, 1H), 4.32 (dd, *J* = 7.7, 3.5 Hz, 1H), 4.17–4.11 (m, 1H), 4.11–4.06 (m, 1H), 4.00 (s, 3H), 3.80–3.71 (m, 2H), 3.64–3.57 (m, 12H), 3.48 (t, *J* = 5.6 Hz, 2H), 3.31 (s, 3H), 3.26–3.18 (m, 3H), 2.81–2.77 (m, 1H), 2.76–2.70 (m, 1H), 2.68–2.62 (m, 1H), 2.07–2.03 (m, 1H), 2.00–1.95 (m, 1H), 1.94–1.87 (m, 1H), 1.80–1.72 (m, 1H), 1.64–1.56 (m, 1H), 1.56–1.50 (m, 1H), 1.47–1.39 (m, 15H), 0.84 (t, *J* = 7.5 Hz, 3H). ^13^C NMR (101 MHz, CD_3_OD) δ 171.9, 169.1, 165.2, 156.3, 153.3, 148.5, 139.8, 134.8, 127.5, 121.4, 117.4, 117.0, 110.1, 80.0, 71.6, 71.5 (3C), 71.4 (3C), 71.2, 71.0, 68.6, 67.6, 59.4, 56.6, 49.7, 48.7, 46.3, 42.2, 41.3, 34.5, 33.4, 32.4, 28.8, 28.6, 21.5, 20.0, 9.1. LC-MS (ESI) *m/z* calc. for C_41_H_65_N_8_O_10_ [M + H]^+^: 829.5, found 829.4.

Tert-butyl (*R*)-(15-(4-(4-((7-ethyl-8-isopropyl-5-methyl-6-oxo-5,6,7,8-tetrahydropteridin-2-yl)-amino)-3-methoxybenzamido)piperidin-1-yl)-15-oxo-3,6,9,12-tetraoxapentadecyl)carbamate (24 mg, 1 eq, 29 μmol) was dissolved to the DCM (1.5 mL) before adding TFA (0.56 mL). The reaction mixture was stirred at room temperature for 3 h and then concentrated to give the desired product (**8**) (21 mg, 25 μmol, 86%). ^1^H NMR (400 MHz, CD_3_OD) δ 7.90 (d, *J* = 8.3 Hz, 1H), 7.56 (s, 1H), 7.51 (d, *J* = 1.9 Hz, 1H), 7.46 (dd, *J* = 8.3, 1.9 Hz, 1H), 4.54–4.41 (m, 3H), 4.14–3.94 (m, 2H), 3.90 (s, 3H), 3.74–3.64 (m, 4H), 3.63–3.52 (m, 12H), 3.22 (s, 3H), 3.21–3.11 (m, 1H), 3.06 (t, *J* = 5.1 Hz, 2H), 2.79–2.57 (m, 3H), 2.09–1.78 (m, 4H), 1.60–1.42 (m, 2H), 1.36 (t, *J* = 7.0 Hz, 6H), 0.78 (t, *J* = 7.4 Hz, 3H). LC-MS (ESI) *m/z* calc. for C_36_H_57_N_8_O_8_ [M + H]^+^: 729.4, found 729.3.

(*R*)-*N*-(1-(1-(5,5-difluoro-7-(1H-pyrrol-2-yl)-5H-5^λ^,6^λ^-dipyrrolo[1,2-c:2’,1’-*f*][1,2,3]diazaborinin-3-yl)-3-oxo-7,10,13,16-tetraoxa-4-azanonadecan-19-oyl)piperidin-4-yl)-4-((7-ethyl-8-isopropyl-5-methyl-6-oxo-5,6,7,8-tetrahydropteridin-2-yl)amino)-3-methoxybenzamide (**11**)

(*R*)-15-(4-(4-((7-ethyl-8-isopropyl-5-methyl-6-oxo-5,6,7,8-tetrahydropteridin-2-yl)amino)-3-methoxybenzamido)piperidin-1-yl)-15-oxo-3,6,9,12-tetraoxapentadecan-1-aminium, trifluoracetate (**8**) (12 mg, 1 eq, 13 μmol) was dissolved in anhydrous DMF (0.4 mL). To the stirred solution, DIPEA was added (8.1 mg, 11 μL, 5 Eq, 63 μmol), followed by BODIPY 576/589 NHS ester (**9**) (5.4 mg, 1 eq, 13 μmol). The reaction mixture was stirred at room temperature for 1 h and then purified via preparative HPLC (column: Phenomenex, Luna 5 μM Phenyl-Hexyl, 100 Å, 75 × 30 mm, 5 micron; mobile phase A: water (0.05% TFA), B: methanol; method: 20–100% B, 6 min + 100% B, 4 min). The desired product (**11**) was obtained as a dark purple solid (9 mg, 9 μmol, 70%). ^1^H NMR (850 MHz, CD_3_OD) δ 7.99 (dd, *J* = 8.4, 2.3 Hz, 1H), 7.60–7.56 (m, 2H), 7.52 (ddd, *J* = 8.4, 4.1, 1.9 Hz, 1H), 7.22–7.19 (m, 2H), 7.19–7.16 (m, 2H), 7.00 (dd, *J* = 4.5, 2.9 Hz, 1H), 6.90 (dd, *J* = 3.9, 1.8 Hz, 1H), 6.36–6.33 (m, 1H), 6.31 (dd, *J* = 4.0, 1.9 Hz, 1H), 4.59–4.52 (m, 2H), 4.52–4.48 (m, 1H), 4.14 (tt, *J* = 11.3, 4.2 Hz, 1H), 4.08–4.04 (m, 1H), 4.00 (s, 3H), 3.77–3.69 (m, 2H), 3.64–3.57 (m, 12H), 3.54 (t, *J* = 5.5 Hz, 2H), 3.39–3.37 (m, 2H), 3.36 (s, 3H), 3.28–3.25 (m, 4H), 3.23–3.17 (m, 1H), 2.81–2.76 (m, 1H), 2.74–2.69 (m, 1H), 2.65–2.60 (m, 2H), 2.11–2.05 (m, 1H), 2.05–2.01 (m, 1H), 2.00–1.95 (m, 1H), 1.94–1.87 (m, 1H), 1.62–1.55 (m, 1H), 1.55–1.48 (m, 1H), 1.45–1.39 (m, 6H), 0.84 (t, *J* = 7.5 Hz, 3H). ^13^C NMR (214 MHz, CD_3_OD) δ 174.8, 172.0, 168.6, 164.4, 156.3, 154.3, 152.1, 150.3, 138.9, 134.9, 133.2, 133.0, 129.8, 127.4, 127.0, 124.9, 124.6, 124.1, 123.2 (2C), 121.0 (2C), 119.1, 117.6, 117.2, 112.3, 111.4, 71.6, 71.5, 71.4, 71.3, 70.6, 68.6, 61.0, 56.8, 53.1, 49.9, 49.5, 49.4, 46.3, 42.2, 40.5, 36.1, 34.5, 33.3, 32.3, 28.9, 28.7, 25.7, 20.6, 19.3, 8.3. LC-MS (ESI) *m/z* calcd. for C_52_H_69_BF_2_N_11_O_9_ [M + H]^+^: 1040.5, found 1040.4. LC-HRMS (ESI) *m/z* calc. for C_52_H_70_BF_2_N_11_O_9_ [M + 2H]^2+^: 520.77041, found 520.77136.

### 4.2. Biology

#### 4.2.1. Cell Culture

HEK293 cells (ATCC) were cultured in DMEM (Gibco) + 10% FBS (Seradigm) with incubation in a humidified 37 °C/5% CO_2_ incubator. N- or C-terminal NLuc/PLK fusions were encoded in pFN31K expression vectors (Promega), including flexible Gly-Ser-Ser-Gly linkers between NLuc and ORFs corresponding to UniProt isoform 1 (PLK1: P53350-1, PLK2: Q9NYY3-1, and PLK3: Q9H4B4-1). Carrier DNA was encoded in pGEM-3Z vectors (Promega). For cellular BRET target engagement experiments, HEK293 cells were transfected with NLuc/PLK fusion constructs using FuGENE HD (Promega) according to the manufacturer’s protocol. Briefly, NLuc/PLK fusion constructs were diluted into Transfection Carrier DNA (Promega) at a mass ratio of 1:9 (mass/mass), after which FuGENE HD was added at a ratio of 1:3 (µg DNA: µL FuGENE HD). One part (vol) of FuGENE HD complexes thus formed was combined with 20 parts (vol) of HEK293 cells suspended at a density of 2 × 10^5^ per mL, followed by incubation in a humidified 37 °C/5% CO_2_ incubator for 20 hr.

#### 4.2.2. NanoBRET assays

##### In-Cell BRET Assays

All BRET assays were performed in white, tissue-culture-treated 96-well plates (Corning #3917) using adherent HEK293 cells at a density of 2 × 10^4^ cells per well. All chemical inhibitors were prepared as concentrated stock solutions in DMSO (Sigma-Aldrich) and diluted in Opti-MEM to prepare working stocks. Cells were equilibrated for 2 h at 37 °C with the appropriate energy transfer probe and test compound prior to BRET measurements, as previously described [15]. Energy transfer probes were prepared at a working concentration of 20× in Tracer dilution buffer (12.5 mM HEPES, 31.25% PEG-400, pH 7.5). Individual kinase NanoBRET assays used the following energy transfer probes and concentrations: PLK1, probe **11** (0.2 µM); PLK2, probe **11** (1.0 µM); PLK3, probe **11** (1.0 µM); WEE1, tracer K10 (Promega, 0.13 µM). To measure BRET, NanoBRET NanoGlo Substrate and Extracellular NLuc Inhibitor (Promega) were added according to the manufacturer’s recommended protocol, and filtered luminescence was measured on a GloMax Discover luminometer equipped with 450 nm BP filter (donor) and 600 nm LP filter (acceptor), using 0.5 s integration time. BRET ratios were calculated by dividing the acceptor luminescence by the donor luminescence. Milli-BRET (mBRET) units (mBU) were calculated by multiplying the raw BRET ratios by 1,000. Broad spectrum profiling on 192 kinases was performed using the K192 NanoBRET Target Engagement assay (Promega) with tracer K10, using the published protocol [24].

Initial evaluation of BRET probes **10** and **11:**

To test BRET probes **10** and **11** against PLK1, HEK293 adherent cells were incubated with either NLuc fused to the N-terminus of PLK1 (NLuc-PLK1) or the C-terminus of PLK1 (PLK1-NLuc) using 1 µM of the BRET probe, and mBRET units were measured. The combination with the highest mBRET units was selected for further processing. NLuc-PLK2 and NLuc-PLK3 were tested with both probes **10** and **11** using the same protocol.

Apparent intracellular potency of BRET probe **11** for NLuc-PLK1:

To estimate the half-maximal effective concentration (EC_50_) of probe **11** to PLK1 in cells, adherent HEK293 cells in a 96-well plate expressing the NLuc-PLK1 fusion protein were mixed with an increasing concentration of BRET probe **11**. The resulting BRET ratio was fitted to the sigmoidal 4-parameter dose–response curve using Prism 9.0 (GraphPad, Boston MA, USA). As a control, the same experiment was repeated in the presence of an excess of unlabeled compound **1** (20 μM) as a competitive inhibitor for 2 h before adding 3× Complete Substrate plus Inhibitor Solution.

Apparent intracellular potency of compound **1** to PLK1, PLK2, and PLK3 using BRET assay:

Adherent HEK293 cells in a 96-well plate expressing NLuc-PLK1, NLuc-PLK2, or NLuc-PLK3 fusion proteins were mixed with increasing concentrations of the BRET probe **11** and various concentrations of unlabeled compound **1** for 2 h before adding 3× Complete Substrate plus Inhibitor Solution. For each probe concentration used, the BRET ratios obtained were used to estimate compound **1** half-maximal inhibitory concentrations (IC_50_) by fitting the data to the four-parameter sigmoidal dose–response curve in Prism 9.0 (GraphPad, Boston MA, USA). To obtain the apparent in-cell binding potency (K_D_ apparent) of **1** towards the PLK1, the linear regression of the estimated IC_50_ versus the concentration of the BRET probe was determined.

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
