# Peer review of "Development of Cell Permeable NanoBRET Probes for the Measurement of PLK1 Target Engagement in Live Cells"

_molecules, 2023, doi:10.3390/molecules28072950_

Round 1

Reviewer 1 Report

The manuscript by Yang, et al describes development of the NanoBRET-based cellular assay that enables measurement of the efficiency of PLK1-3 inhibitors in the cellular milieu. Generally, the manuscript is well-structured and the study is methodologically solid, yet there are some issues that should be addressed prior to acceptance for publication:

1. Figure 2c – is the x-scale really in Molar? This would imply that the authors tested a range from above millimolar to above ten micromolar, which does not go in line with the IC50 values listed below the graph.

2. Reporting of IC50 values with the precision of 3 significant digits (e.g., in Figure 2c) is not justified if the authors are not showing the measurement uncertainty. Standard deviation calculated from the results of independent experiments should also be reported in case of values listed in Table 1.

3. In Figure 2 and Figure 3, are the results from a single independent experiment shown? What do the error bars indicate?

4.      Line 152: should „lowest reported potency for inhibition of cell proliferation“ read as „highest reported potency / lowest reported concentration required for inhibition of cell proliferation“?

5.      Line 162 and Figure 3a: please provide the literature reference for the maximal plasma concentration of adavosertib. In https://www.ncbi.nlm.nih.gov/pmc/articles/PMC7338825/ , higher Cmax is reported (over 1 microM).

6.      Lines 171-173: the authors first state that the inhibitor shows target engagement and later that the targets show target engagement. The second sentence should be rephrased for better clarity.

7.      Under discussion, the authors mention ATP levels as the factor dictating the shift of intracellular IC50 of inhibitors relative to the enzymatic assays (lines 181-184); however, the expression level of enzyme might be just as important (quite obviously, the IC50 of a compound cannot be lower than half of the concentration of the enzyme in cells). Have the authors compared the levels of endogenous vs sensor-encoded PLK1, e.g., using a PLK1-selective antibody on the Western blot?

8.      Line 188: is “drop off” correct? Just “drop” or “decline” sounds more academic.

9.      Line 213 – should “select” read as “selective”?

10.   Please discuss: PLK1 is not catalytically active by default (e.g., see https://pubmed.ncbi.nlm.nih.gov/28591578/). How can the absence of activating factors affect the results of the developed assay?

11.   Line 232: „All chemicals were commercially available except those whose synthesis is described“ – I think that at least the sources of PLK inhibitors used should be clearly listed.

12.   Materials and methods section: theoretical calculated m/z values should be added in brackets behind the experimentally measured ESI-MS data.

13.   Materials and methods section: the cloning procedures should be described in more detail – e.g., the authors should mention whether the PLK1/PLK2/PLK3 DNA-sequences were synthesized or PCR-amplified from any kind of template; primers and restrictases and the sources thereof should also be clearly named. While I understand that cloning might be a routine procedure for the authors from Promega, it should be made sufficiently transparent to the readers.

14.   Did the authors optimize the incubation time of cells with the probe and the competing compounds? 2 h seems relevant, yet in case if viability of the cells is compromised by the presence of high concentrations of inhibitors, it can lead to significant measurement error (e.g., due to the compromised integrity of cell plasma membrane). Did the authors measure viability of cells or check the cells for the phenotype changes that would reflect apoptotic behaviour? If any illustrative material regarding the cellular viability/phenotype is available, it should be shown as a part of the Supplementary Information.

Reviewer 2 Report

Th manuscript by Yang Xuan and Colleagues describes the synthesis and initial bioassay of one energy transfer probe for detecting Polo-Like Kinase 1 target engagement in living cell cultures. 

The paper is well written and supported by figures. Experiments can be followed quite well even if the text could be more explicative, as detailed below. I have no major issues concerning the chemical synthesis, that appears complete. However, I think the biological assays should be better described in order to understand the relevance of results. My concerns are:

1. line 95: Authors write that HEK293T cells were used for titration, while in the rest of the paper KEK 293 cells are mentioned: since these are two different cell lines (293T derives from the other after transfection with simian virus 40), with different biological properties, the Author must disclose if they used 2 different cell lines and why they did it.

2. In fig. 2, specify how many tests were conducted, and if they were technical or biological replicates.

3. Table 1 is misleading for two reasons. First, it is not clear what are the data obtained by the authors and what are from references from other groups. Second, and most important, it makes no biological sense to compare results obtained from different cell lines, given that some are cancerous and others not. Please make these informations apparent in the table and in the discussion. Lastly, what's the meaning of 'data is an average of n=2'? Technical or biological replicates? Are 2 replicates sufficient? What is their variation? With 3 replicates it would be possible to add a measure of variation (mean or median +/- confidence interval or standard deviation or SEM...). At present, the table does not show robust data that we can directly compare.

4. Why the WEE1 kinase inhibitor was not tested also on pancreatic cancer cells-apparently it was tested on human embryonic kidney only.

5. Methods. Line 361, specify how many passages did the cells before testing. Line 384, why milliBRET? If you multiply by 1000, it should be KiloBRET, like Kilometers (1000 meters), millimeters (1 meter divided by 1000). Lines 399 and 404: incubations for 2 hours, specify at what temperature.

MINOR POINTS:

Line 128: 'PLK2and PLK2' ? Is this correct?

Line 130: 'figure 1a' I guess it is 'figure 2a'

Line 185: please add the reference for PLK2 and PLK3.

Line 228: delete one fullstop.

Round 2

Reviewer 2 Report

Authors have adequately solved by concerns.